# Distinct functions of three Wnt proteins control mirror-symmetric organogenesis in the *C. elegans* gonad

**Shuhei So[1][†], Masayo Asakawa[1], Hitoshi Sawa[1,2]***

[1]Multicellular Organization Laboratory, Department of Gene Function and Phenomics, National Institute of Genetics, Research Organization of Information and Systems (ROIS), Mishima, Japan; [2]Department of Genetics, School of Life Science, SOKENDAI (The Graduate University for Advanced Studies), Mishima, Japan

**Abstract** Organogenesis requires the proper production of diverse cell types and their positioning/migration. However, the coordination of these processes during development remains poorly understood. The gonad in *C. elegans* exhibits a mirror-symmetric structure guided by the migration of distal tip cells (DTCs), which result from asymmetric divisions of somatic gonadal precursors (SGPs; Z1 and Z4). We found that the polarity of Z1 and Z4, which possess mirror-symmetric orientation, is controlled by the redundant functions of the LIN-17/Frizzled receptor and three Wnt proteins (CWN-1, CWN-2, and EGL-20) with distinct functions. In *lin-17* mutants, CWN-2 promotes normal polarity in both Z1 and Z4, while CWN-1 promotes reverse and normal polarity in Z1 and Z4, respectively. In contrast, EGL-20 inhibits the polarization of both Z1 and Z4. In *lin-17 egl-20 cwn-2* triple mutants with a polarity reversal of Z1, DTCs from Z1 frequently miss-migrate to the posterior side. Our further analysis demonstrates that the mis-positioning of DTCs in the gonad due to the polarity reversal of Z1 leads to mis-migration. Similar mis-migration was also observed in *cki-1(RNAi)* animals producing ectopic DTCs. These results highlight the role of Wnt signaling in coordinating the production and migration of DTCs to establish a mirror-symmetric organ.

**\*For correspondence:**
hisawa@nig.ac.jp

**Present address:** [†]Department of Reproductive and Perinatal Medicine, Hamamatsu University School of Medicine, Hamamatsu, Japan

**Competing interest:** The authors declare that no competing interests exist.

## Editor's evaluation

The Wnt signaling pathway controls cell polarity, although the role of Wnts themselves remains controversial. This important study addresses the involvement of the Wnt pathway in the specification and migration of *C. elegans* DTCs and shows how they are symmetrically polarized. The paper presents compelling evidence that will be of interest for understanding a striking example of cell polarity and, for the *C. elegans* community, to show that DTCs can migrate independently of the germline.

## Introduction

Most animals belonging to Bilateria have a mirror-symmetric body plan along the left-right axis, except for most internal organs. In mirror-symmetric tissues, cells in equivalent positions should acquire the same cell fates. Additionally, they are likely to exhibit mirror-symmetric polarity orientation for mirror-symmetric morphogenesis. However, the mechanism to produce mirror symmetric polarity has not even been postulated except for a few examples. At the midline of the neural rod in the developing zebrafish neural tube, the polarity regulator Pard3 is localized on the cleavage furrow of neural progenitors and inherited by the two daughter cells on either side of the midline (*Tawk et al., 2007*). This results in a mirror-image apico-basal polarity of the daughter cells and the subsequent formation

**eLife digest** In humans and other animals, cells are organized into tissues and organs that each perform distinct roles in the body. Some organs and tissues have a mirror-symmetric structure, meaning they are divided into two halves that are exact reflections of one another. However, it is not fully understood how these types of structures form during development.

The formation of mirror-symmetric structures often relies on cell polarity, which is when the components of a cell – such as its structure, internal contents and functional regions – are unevenly distributed. In the nematode worm *C. elegans*, for example, their mirror-symmetric gonads (or sex organs) are formed by two polarized cells called Z1 and Z4.

Both Z1 and Z4 divide asymmetrically to produce two daughter cells with distinct concentrations of a particular transcription factor. For Z1, the daughter cell facing the anterior of the gonad has lower levels of the transcription factor than the posterior daughter cell, while the two cells generated by Z4 have the opposing mirror asymmetry. This polarity drives the production of two distal tip cells – one produced by the anterior daughter cell of Z1 and the other by the posterior daughter cell of Z4 – which migrate to opposite ends of the gonad.

A cell signaling pathway known as Wnt is crucial for establishing cell polarity in many species. However, a previous study found that *C. elegans* could still develop healthy gonads even when all five ligand proteins that activate the Wnt pathway were mutated. Here, So et al. reveal that these mutations can impact polarity, but only when LIN-17, the receptor for the Wnt ligands, is also mutated. Further experiments showed that LIN-17 can independently regulate cell polarity and compensate for the loss of Wnt signaling.

So et al. also identified three specific Wnt ligands – CWN-1, CWN-2 and EGL-20 – that collectively control the polarity of Z1 and Z4. Each protein has a distinct role: CWN-1 promotes Z1 and Z4 to have the same polarity, while CWN-2 induces the polarity of Z1 cells to reverse. EGL-20 then stops Z1 from regaining its original polarity and no longer mirroring the polarity of Z4.

These findings shed new light on how Wnt signaling contributes to the mirror-symmetric structure of *C. elegans* gonads. It is possible that these proteins play similar roles in other animals to help regulate how organs form.

of the ventricle. In the dorsal and ventral sides of the compound eye of *Drosophila*, ommatidia have mirror-image chirality that is regulated by protocadherins Fat/Dachsous and PCP (planar cell polarity) signaling (*Rawls et al., 2002*).

During the development of *C. elegans*, most cells are polarized in the same anterior-posterior orientation and divide asymmetrically to produce distinct daughter cells (*Sawa, 2012*). This polarity is regulated by the Wnt signaling pathway known as the Wnt/β-catenin asymmetry pathway (*Mizumoto and Sawa, 2007*). Nuclear localization of POP-1/TCF, for example, is higher in the anterior than the posterior daughter cells (hereafter called HL polarity for high-low POP-1 concentration). However, there are some exceptions to the rule that result in mirror-symmetric polarity. During the development of the vulva, one of the vulval precursor cells (VPCs) P7.p has reversed polarity with higher POP-1 in the posterior daughter (LH polarity for low-high POP-1 concentration) (*Deshpande et al., 2005*). The mirror symmetric polarity between P7.p and another VPC P5.p is essential for their mirror-symmetric lineages and the structure of the vulva. The LH polarity of P7.p is instructed by Wnt proteins secreted from the anchor cell located between P5.p and P7.p (*Green et al., 2008*). In the gonad, at the L1 stage, somatic gonadal precursor (SGP) cells, Z1 and Z4 have LH and HL polarity, respectively, creating their mirror-symmetric lineages producing distal tip cells (DTCs) from the distal daughters (Z1.a and Z4.p) (*Siegfried et al., 2004*; *Figure 1A and B*). However, it is not known how this mirror-symmetric polarity is established. We have shown previously that SGP polarity is not affected in quintuple Wnt mutants that have mutations in all five Wnt genes in *C. elegans*, suggesting that Wnts may not be required for SGP polarity (*Yamamoto et al., 2011*).

DTCs, which are the most distal granddaughters of SGPs (Z1.aa and Z4.pp), have two functions. First, DTCs function as niche cells for germline stem cells, inhibiting their entry into meiosis by expressing the Notch ligand LAG-2 (*Henderson et al., 1994*). Additionally, during gonadogenesis, each DTC migrates with a U-shaped trajectory to guide extension of gonad arms, resulting in

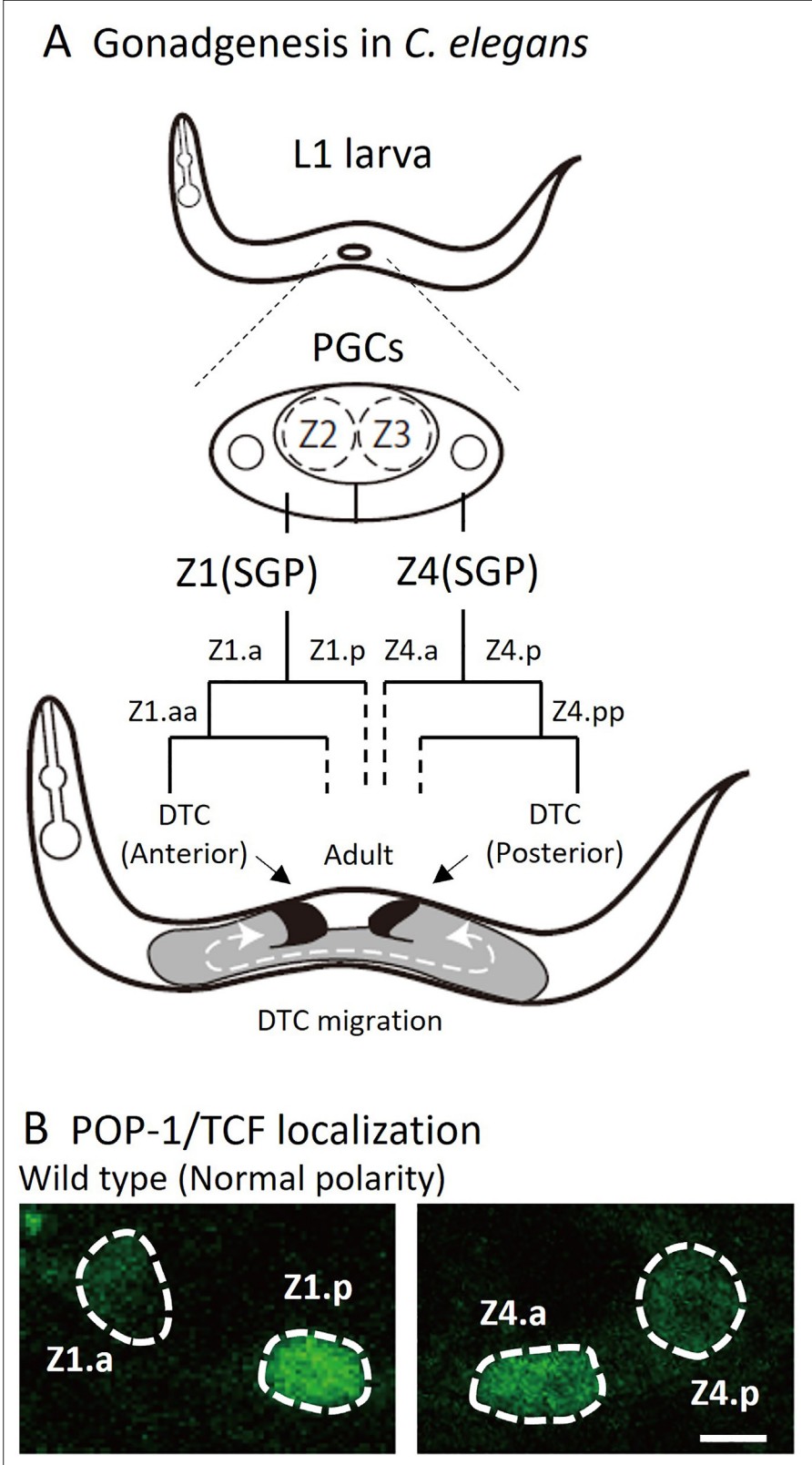

**Figure 1.** Cell lineage of somatic gonad and asymmetric localizations of POP-1/TCF between somatic gonadal precursor (SGP) daughter cells. (**A**) After hatching, L1 animals have two SGPs (Z1 and Z4) and primordium germ cells (Z2 and Z3). SGPs undergo mirror-symmetric divisions along the proximal-distal axis, and their distal granddaughters Z1.aa and Z4.pp become distal tip cells (DTCs) that migrate anteriorly and posteriorly, respectively.

*Figure 1 continued on next page*

*Figure 1 continued*

The U-shape of gonad arms is established through the migration of DTCs. (**B**) Examples of POP-1 localizations in wild-type animals using GFP::POP-1 (*qIs74*). Animals were cultured at 22.5°C. Anterior is to the left. Scale bars indicate 2 μm.

U-shaped gonad both on the anterior and posterior sides (*Figure 1A*). A recent report suggests that DTCs may not actively migrate by themselves; instead, they are pushed distally by proliferating germ cells (*Agarwal et al., 2022*).

While the quintuple Wnt mutants exhibit normal SGP polarity, we observed a significant impact on polarity when Wnt mutations are present in the background of the *lin-17*/Frizzled mutation. For example, triple Wnt mutations (*cwn-1 cwn-2 and egl-20*) combined with the *lin-17* mutation disrupt the polarity of both Z1 and Z4. LIN-17 functions in a Wnt-independent manner, as LIN-17 lacking the Wnt binding domain (cysteine-rich domain [CRD]) can rescue gonadal defects in compound mutants containing the *lin-17* mutation. In the *lin-17* background, the three Wnts have distinct functions. Specifically, *cwn-1* promotes HL polarity in both Z1 and Z4, while *cwn-2* promotes LH and HL polarity in Z1 and Z4, respectively. In contrast, *egl-20* inhibits HL polarity induced by *cwn-1*. In *lin-17; egl-20 cwn-2* animals, both Z1 and Z4 show HL polarity disrupting the mirror symmetry of polarity. Notably, in this genotype, we observed that the DTC from Z1 frequently migrates posteriorly similar to that from Z4. We further demonstrated that the ectopic positions of DTCs in the center of the gonad cause mis-migration. Our results suggest that the distal positions of DTCs in the gonad through the mirror-symmetric polarity of SGPs are required for stable distal migration, consistent with the recent report supporting the permissive migration model (*Agarwal et al., 2022*). However, the posterior migration of ectopically positioned DTCs from Z1 passing through germ cells in the triple mutants, along with DTC migration in germless *mes-1* mutants, strongly suggests self-migratory mechanisms of DTCs.

## Results
### DTC production is controlled by redundant functions of Wnts and LIN-17/Frizzled

We have previously demonstrated that DTC production and SGP polarity remain normal in any Wnt single or compound mutant strains including quintuple Wnt mutants (*lin-44; cwn-1; egl-20 cwn-2; mom-2*), suggesting that Wnts may not be required for SGP polarity (*Yamamoto et al., 2011*). However, when a *cwn-2* mutation was combined with that of *lin-17*/Frizzled, we observed that most animals lacked one or both DTCs (a missing DTC phenotype) as determined by the expression of a DTC marker, *mig-24*::Venus (*Tamai and Nishiwaki, 2007*). While *lin-17*(*n3091* with nonsense mutation) single mutants exhibit a weak missing DTC phenotype (*Table 1*; Sternberg and Horvitz, 1988), *lin-17; cwn-2* showed a strong enhancement in regard to the absence of the anterior DTC (*Table 1*: p<0.0001 by Pearson's chi-square test). We observed a similar phenotype in *lin-17*(*n671*)*; cwn-2* double mutants, confirming that this genetic interaction is not allele-specific. Although mutations in other Wnt genes (*cwn-1* and *egl-20*) by themselves or in combination did not cause such enhancements, the *cwn-1* mutation enhanced the phenotype of *lin-17; cwn-2* for both the anterior and posterior DTCs (p<0.0001 by Pearson's chi-square test), resulting in the majority of *lin-17; cwn-1; cwn-2* triple mutants lacking both gonadal arms and being sterile. These results indicate that *lin-17*/Frizzled and Wnt genes (*cwn-1* and *cwn-2*) regulate DTC production through parallel pathways.

We note that the DTC defects in various compound mutants described above and below were more severe at 15°C compared to 22.5°C for unknown reasons. Since the temperature sensitivities of individual mutations have not been reported and most mutations are nonsense or deletions, we hypothesize that the processes for DTC production are somehow cold-sensitive.

We found that *lin-17; egl-20 cwn-2* triple mutants exhibited a unique phenotype that has not been reported before. The triple mutants frequently displayed two DTCs in the posterior region accompanied by the absence of a DTC in the anterior region (this phenotype is hereafter referred to as Dpd for double posterior DTCs) (*Figure 2*). We have not observed triple mutants with three DTCs. These observations suggest that the Dpd phenotype is caused by the miss-migration of the anterior DTC derived from the Z1 cell, rather than the extra production of posterior DTCs from the Z4 cell.

**Table 1.** Missing distal tip cell (DTC) phenotype of compound mutants.

| | Genotype | Anterior | Posterior | n |
|---|---|---|---|---|
| WT | N2 | 0.0% | 0.0% | 100 |
| Wnt receptors | *lin-17(n3091)** | 4.7% | 2.8% | 107 |
| | *lin-17* at 15°C* | 11.1% | 1.9% | 54 |
| | *mom-5*§ | 0.0% | 0.0% | 54 |
| | *lin-17 mom-5*§ | 100% | 100% | 30 |
| | +*osIs113* (ΔCRD-LIN-17)*, § | 0.0% | 0.0% | 34 |
| | *lin-17(os2) mom-5*§ | 2.0% | 1.0% | 100 |
| | *lin-17(mn589) mom-5*§ | 0.0% | 0.0% | 50 |
| | *mig-1 lin-17(n671)* | 9.0% | 9.0% | 67 |
| | *lin-17; cfz-2* | 9.4% | 4.7% | 64 |
| | *lin-17; lin-18*§ | 7.6% | 3.8% | 53 |
| | *lin-17; cam-1*† | 5.7% | 0.0% | 300 |
| | | | | |
| Fz+Wnt | *lin-17; cwn-1*† | 7.2% | 0.9% | 111 |
| | *lin-17; cwn-2** | 45.8% | 4.3% | 94 |
| | +*osIs113* (ΔCRD-LIN-17)* | 0.0% | 0.0% | 40 |
| | *lin-17; cwn-2* at 15°C* | 88.5% | 54.1% | 61 |
| | +*osIs113* (ΔCRD-LIN-17) at 15°C* | 0.0% | 0.0% | 45 |
| | *lin-17(n671); cwn-2** | 48.8% | 7.3% | 41 |
| | *lin-17; egl-20*† | 4.4% | 0.0% | 340 |
| | *lin-17; egl-20 cwn-2*\*, ‡ | 2.9% | 0.0% | 70 |
| | *lin-17; egl-20 cwn-2* at 15°C\*, ‡ | 0.3% | 3.0% | 332 |
| | *lin-17; cwn-1; egl-20** | 5.8% | 1.5% | 69 |
| | *lin-17; cwn-1; egl-20* at 15°C* | 1.3% | 1.3% | 78 |
| | *lin-17; cwn-1; cwn-2*\*, § | 89.1% | 78.3% | 46 |
| | *lin-17; cwn-1; cwn-2* at 15 °C\*, § | 100% | 97.6% | 42 |
| | *lin-17; cwn-1; egl-20 cwn-2*\*, § | 86.8% | 84.9% | 53 |
| | *lin-17; cwn-1; egl-20 cwn-2* at 15°C\*, § | 97.8% | 95.7% | 46 |
| | +*osEx395* (*ceh-22p*::CWN-1::Venus) at 15°C\*, § | 96.0% | 100% | 25 |
| | +*osIs93* (*egl-20p*::CWN-2::Venus) at 15°C\*, § | 88.9% | 74.1% | 27 |
| | *mom-5; cwn-1; egl-20 cwn-2*†, § | 0.0% | 0.0% | 22 |

n3091 was used as a *lin-17* mutation unless otherwise indicated.

The strains were grown at 22.5°C unless otherwise indicated.

*The strains had *tkIs12* encoding the *mig-24p*::Venus that expresses in DTCs.

†The strain had *vpIs1* encoding *elt-3*::GFP that was used to observe hypodermal defects.

‡Dpd animals were considered to have both anterior and posterior DTCs.

§Homozygous progeny from balanced heterozygotes (see **Supplementary file 1**).

n: number of animals scored.

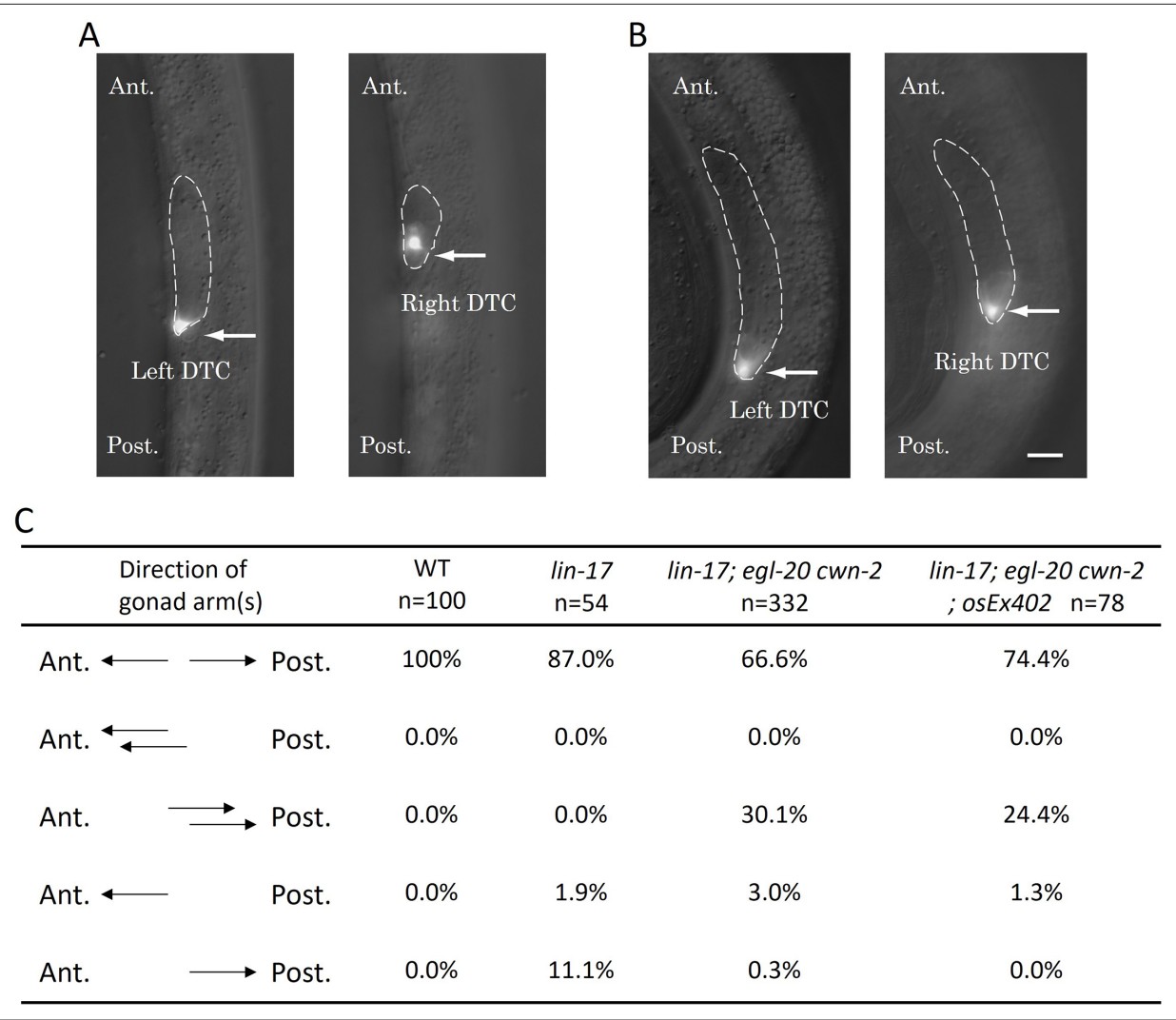

**Figure 2.** The Dpd phenotype of *lin-17; egl-20 cwn-2* animals. Merged images of DIC and GFP of *lin-17; egl-20 cwn-2* animals at the L2 stage (**A**) and the L3 stage (**B**) are presented. Arrows highlight Z4-derived left and Z1-derived right distal tip cells (DTCs) expressing *mig-24*::Venus. The gonad is delineated with dashed lines. The anterior is up. Left and right images in each panel are correspond to the same area with different focal planes in the same animals. (**C**) The table compiles the direction of gonad arms extension. These experiments were conducted with animals grown at 15°C. Ant. and Post. indicate anterior and posterior, respectively. Scale bars indicate 10 μm.

More analyses of the mis-migration phenotype are shown below. In *Table 1*, we considered that Dpd animals have both anterior (Z1-derived) and posterior (Z4-derived) DTCs.

*lin-17; cwn-1; egl-20 cwn-2* quadruple mutants exhibited a similar phenotype to *lin-17; cwn-1; cwn-2* triple mutants with most animals lacking both DTCs. However, the Dpd phenotype was not observed in the quadruple mutants.

## LIN-17 functions in a Wnt-independent manner for the DTC production

In contrast to the pronounced DTC defects observed in *lin-17*+Wnt compound mutants as described above, our previous findings demonstrated that quintuple Wnt mutants, where all five Wnt genes are mutated, exhibit normal DTC production (*Yamamoto et al., 2011*). This suggests that LIN-17 alone is sufficient for DTC production in the absence of Wnts and that LIN-17's function is Wnt-independent. To confirm the Wnt-independent role of LIN-17, we constructed LIN-17 with a deletion of the CRD, which is the Wnt binding domain, and expressed it under the *lin-17* promoter (*osIs113*; ΔCRD-LIN-17). As expected, the missing DTC phenotype of *lin-17; cwn-2* and *lin-17 mom-5*/Frizzled was completely rescued by *osIs113* (*Table 1*). Consistently, *lin-17* with missense mutations of the conserved cysteine

residues of CRD (*os2* and *mn589*) combined with the *mom-5* mutation showed only minor, if any, DTC defects (*Table 1*), in contrast to the complete loss of DTCs in *lin-17(n3091) mom-5*. These results strongly support the conclusion that LIN-17 regulates the DTC production in a Wnt-independent manner.

The results also suggest that MOM-5/Frizzled might be the receptor for Wnts regulating DTC production, as *lin-17 mom-5* double mutants completely lack DTCs. Combining mutations in the other known Wnt receptor genes (*mig-1*/Frizzled, *cfz-2*/Frizzled, *cam-1*/Ror, and *lin-18*/Derailed) with that of *lin-17* did not significantly affect the gonadal phenotype (*Table 1*). Additionally, *mom-5* combined with triple Wnt mutations did not induce the missing DTC phenotype in the absence of *lin-17* mutations. While other receptors might also play some roles, MOM-5 appears to be a major receptor for Wnts regulating DTC production.

## SGP polarity is redundantly regulated by LIN-17 and multiple Wnts

It has been reported that the absence of DTCs in mutants of intracellular components of the Wnt signaling pathway (Wnt/β-catenin asymmetry pathway) is caused by the loss of polarity of SGPs (Z1 and Z4) which is required for their asymmetric divisions (*Siegfried et al., 2004*). To determine whether compound mutants of *lin-17* and Wnt mutations also affect SGP polarity, we analyzed the localization of *sys-1*p::GFP::POP-1 (*qIs74*) after SGP divisions (*Figures 1 and 3*). In the wild-type, POP-1 is preferentially localized to the nuclei of proximal SGP daughters (Z1.p and Z4.a) compared to the distal ones (Z1.a and Z4.p), representing SGP polarity (LH and HL polarity in terms of *sys-1*p::GFP::POP-1 localizations for Z1 and Z4, respectively). These asymmetries were strongly disrupted and weakly affected in *lin-17 mom-5* double and *lin-17* single mutants, respectively, as described previously (*Phillips et al., 2007*; *Siegfried et al., 2004*). In contrast, our previous finding demonstrated that SGP polarity was normal in quintuple Wnt mutants (*lin-44; cwn-1; egl-20 cwn-2; mom-2*) (*Yamamoto et al., 2011*).

To evaluate SGP polarity using *sys-1p*::GFP::POP-1, we first quantified the ratios of signal intensities (on a logarithmic scale) of *sys-1p*::GFP (NLS) which localizes symmetrically between the daughter cells, and calculated the 95% confidence interval (CI) for symmetrically localized signals. Then, we quantified the ratios (on a logarithmic scale) of *sys-1p*::GFP::POP-1 signal intensities proximal to distal daughter cells in various genotypes (*Figure 3A* and *Figure 3—figure supplement 1*). Values within the 95% CI (between the red lines in *Figure 3B*) indicate symmetric localization, while values lower than the 95% CI (below the lower red line) indicate reversed localization, respectively. Most compound mutants containing *lin-17* and *cwn-2* mutations exhibited symmetric localization except for *lin-17; egl-20 cwn-2*, consistent with their missing DTC phenotype (*Table 1*). The results strongly suggest that the missing DTC phenotype of these mutants is caused by symmetric divisions of SGPs. On the other hand, *lin-17; egl-20 cwn-2* mutants that exhibited the Dpd phenotype showed reversed Z1 polarity, suggesting a possible link between the polarity reversal of Z1 and the Dpd phenotype (see below).

To understand the effects of each mutation on SGP polarity, we next statistically compared the distributions of the POP-1 signal ratios among the compound mutants (statistical analyses in this chapter were done by Student's t-test). To evaluate the effects on SGP polarization irrespective of orientation, we compared the absolute ratios (on a logarithmic scale) of signal intensities between proximal and distal daughter cells (*Figure 3—figure supplement 2*) rather than the signed ratios described in *Figure 3B*. Symmetric POP-1 localization in the *lin-17* mutant was strongly and moderately enhanced by the *cwn-2* mutation for the Z1 and Z4 cells, respectively (p<0.0001 for Z1 and p=0.0108 for Z4 at 15°C; p=0.1400 for Z1 and p=0.4764 for Z4 at 22.5°C in comparison of absolute differences). In the *lin-17; cwn-2* background, this symmetry was further enhanced by the *cwn-1* mutation for the Z4 cell (*Figure 3—figure supplement 2*: p=0.0995 at 15°C; p=0.0088 at 22.5°C in comparison of absolute differences), indicating that, in the absence of LIN-17, Z1 polarity is mostly controlled by CWN-2, while Z4 polarity is regulated by both CWN-1 and CWN-2. In contrast to the effects of *cwn-1*, the *egl-20* mutation rescued the polarization defect in the *lin-17; cwn-2* background at least at 15°C (p<0.0001 for Z1 and p=0.0007 for Z4 in comparison of absolute differences), indicating that *egl-20* has a negative role for SGP polarization. SGP polarization in *lin-17; egl-20 cwn-2* appears to depend on *cwn-1* at 22.5°C (p=0.0592 for Z1 and p<0.0001 for Z4 in comparison of absolute differences).

In contrast to the rescue of polarization by *egl-20* in the *lin-17 cwn-2* background, comparison of signed but not absolute differences (*Figure 3B*) showed it causes reversal of Z1 but not Z4 polarity (p<0.0001 at 15°C and p=0.0015 at 22.5°C). Interestingly, in this triple mutant, especially at 15°C, Z1

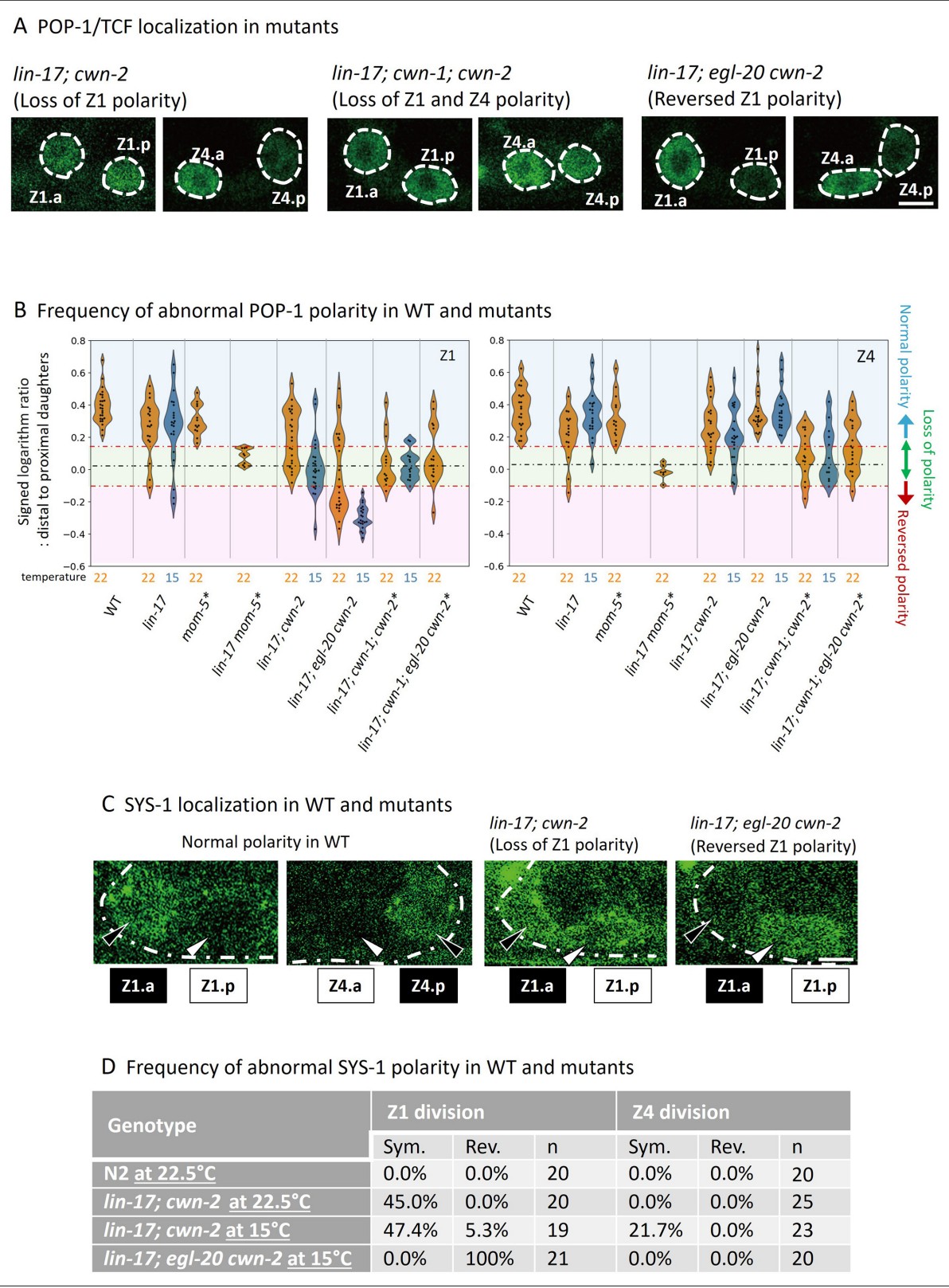

**Figure 3.** POP-1 and SYS-1 asymmetry is redundantly regulated by *lin-17* and multiple Wnts. (**A**) Examples of *sys-1p*::GFP::POP-1 (*qIs74*) localizations in the indicated genotypes. Animals were cultured at 22.5°C, except for *lin-17; egl-20 cwn-2*, which was cultured at 15°C. (**B**) Violin plots illustrate the distribution of signed ratios on a logarithmic scale of GFP::POP-1 signals proximal to distal daughters. The violin plots show the distribution of experimental data at two temperatures 22.5°C (22 in the figure) and 15°C, represented on the left and right halves of each violin, respectively. The

*Figure 3 continued on next page*

*Figure 3 continued*

black dashed lines represent the zero residual lines (where the predicted values equal the observed values), and the red dashed lines indicate the 95% confidence interval (CI) calculated from signals of symmetrically localizing *sys-1p*::GFP(NLS). Values within the 95% CI (between the red lines; light green regions) indicate symmetric localization. Values below the lower red line (light blue regions) indicate reversed localization, while values above the upper red line (light red regions) indicate normal localization. See Materials and methods for details. For the strains indicated by asterisks, homozygous progeny from balanced heterozygotes (see *Supplementary file 1*) was analyzed. (C) Examples of SYS-1 localizations in the indicated genotypes were observed using VENUS::SYS-1 (*qIs95*). Animals were cultured at 22.5°C, except for *lin-17; egl-20 cwn-2*, which was cultured at 15°C. (D) Abnormal SYS-1 localization in compound mutants. SYS-1 localizations were analyzed using *qIs95* (Venus::SYS-1). Sym: the fluorescence was observed in both daughter cells. Rev: the fluorescence was observed in the proximal daughter cells. n: number of animals scored. Since somatic gonadal precursor (SGP) daughter cells are often present at distinct focal planes, we normalized the depth effects on fluorescence intensities (see Materials and methods for details) for the quantification shown in (B). The images in (A) and (C) are from animals with SGP daughters at similar depths. Scale bars indicate 2 μm. Source data is available (*Figure 3—source data 1*).

The online version of this article includes the following source data and figure supplement(s) for figure 3:

**Source data 1.** Data utilized to generate the graph in *Figure 3B and D*.

**Figure supplement 1.** Examples of POP-1 localizations in the indicated genotypes observed using *sys-1p*::GFP::POP-1 (*qIs74*).

**Figure supplement 2.** Comparison of absolute differences in POP-1 asymmetry regulated by *lin-17* and multiple Wnts.

**Figure supplement 3.** CWN-1 expression is not affected in *lin-17; egl-20 cwn-2* animals.

and Z4 have the same HL polarity orientation, indicating that the mirror symmetry of SGP polarity is established through the functions of *lin-17*, *egl-20,* and *cwn-2*.

Although the *egl-20* mutation suppressed the loss of polarity phenotype in the *lin-17; cwn-2* background, it did not in the *lin-17; cwn-1 cwn-2* background (p=0.6327 and 0.5741 for symmetric Z1 and Z4 polarity, respectively, between *lin-17; cwn-1; cwn-2* and *lin-17; cwn-1; egl-20 cwn-2* in comparison of absolute differences at 22.5°C: *Figure 3—figure supplement 2*), suggesting that *egl-20* represses *cwn-1* function. One possible explanation may be that *egl-20* suppresses the expression of *cwn-1*. However, the expression levels of *cwn-1p*::CWN-1::GFP (*Yamamoto et al., 2011*) are similar among wild-type, *lin-17; cwn-2*, and *lin-17; egl-20 cwn-2* animals (*Figure 3—figure supplement 3*).

To further confirm the defects in SGP polarity in the compound mutants, we also analyzed the localization of SYS-1/β-catenin (*Figure 3C and D*) which is regulated independently of POP-1 localization (*Phillips et al., 2007*). In both wild-type and *lin-17* single mutants, Venus::SYS-1 is preferentially localized to the distal SGP daughters than the proximal ones (*Phillips et al., 2007*). Consistent with the defects of POP-1 localization, we found that *lin-17; cwn-2* exhibited symmetric SYS-1 localization for Z1 and less frequently for Z4, while it was reversed for Z1 in *lin-17; egl-20 cwn-2* (*Figure 3C and D*). These results demonstrated that SGP polarity is regulated by the parallel functions of LIN-17/Fz and Wnts.

To understand how SGP polarity orientation is regulated, we examined whether Wnt functions are instructive or permissive. We attempted to reverse Wnt gradients by ectopically expressing CWN-1 and CWN-2 which are normally expressed posteriorly and anteriorly relative to the gonad, respectively (*Harterink et al., 2011*). We utilized *osEx395* (*ceh-22p*::CWN-1::Venus expressed in the pharynx) and *osEx402* (*egl-20p*::CWN-2::Venus expressed near the anus) which have been demonstrated to provide weak and strong rescue, respectively, of polarity defects in seam cells in the *cwn-1; egl-20 cwn-2* background (*Yamamoto et al., 2011*). However, these extrachromosomal arrays failed to significantly affect the production of DTCs in *lin-17; cwn-1; egl-20 cwn-2* (instead of *osEx402*, its integrant *osIs93* was used) (*Table 1*) nor mis-migration of DTCs in *lin-17; egl-20 cwn-2* for *osEx402* (*Figure 2*). Therefore, it remains unclear whether these Wnts are instructive or permissive for SGP polarity.

## Ectopically produced DTCs from Z1 can migrate posteriorly

In the wild-type, the anterior and posterior DTCs are born and migrate on the right-anterior and left-posterior sides of the body, respectively. In *lin-17; egl-20 cwn-2* mutants with the Dpd phenotype at the early L3 stage, the more posterior DTC among the two DTCs was observed on the left side (left-DTC) at the tip of the gonadal arm similar to the posterior DTC in the wild-type. Meanwhile, the more anterior DTC was on the right side (right-DTC) at the end of the right side gonad. The cup shape of this right-DTC with the reversed orientation compared to that in the right-anterior DTC in wild-type suggests that it was migrating toward the posterior side (*Figure 2B*). At the late L3 and early L4

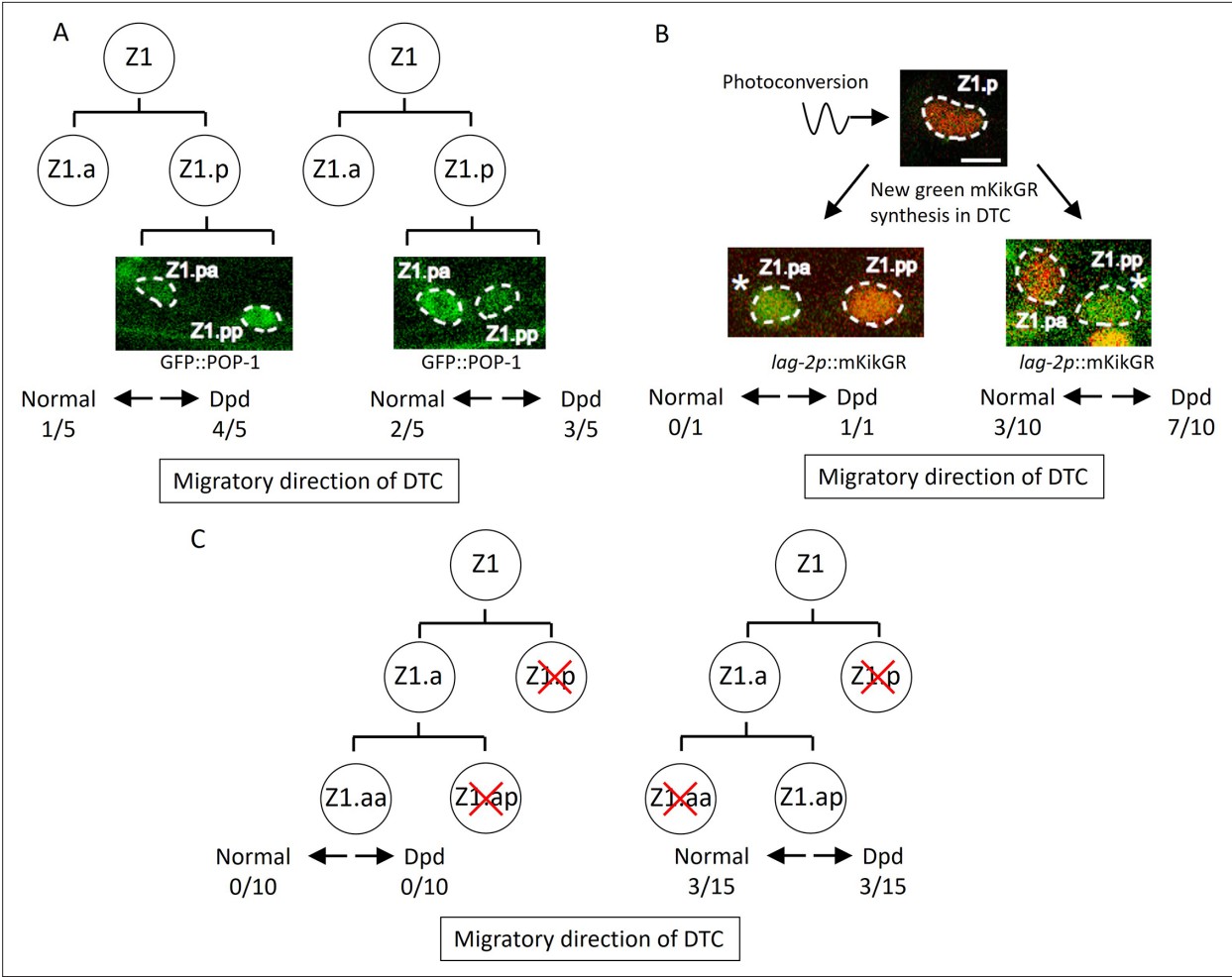

**Figure 4.** Abnormal positions of distal tip cells (DTCs) in *lin-17; cwn-2 egl-20* mutant cause the Dpd phenotype. (**A**) POP-1 asymmetry in Z1.p daughter cells was observed at the late L1 stage. LH (normal) POP-1 polarity (left panel) and HL (reversed) POP-1 polarity (right panel) indicate that Z1.pa and Z1.pp become DTCs, respectively. After the recovery and growth of the animals, the migratory directions of DTCs were evaluated at the L4 stage by extending gonad arms. (**B**) Green fluorescence of *lag-2p*::mkikGR (*osIs168*) in Z1.p of *lin-17; cwn-2 egl-20* animals was photoconverted to red by irradiation with a 405 nm light laser. After the recovery and 10 hr of growth of the animals, Z1.p daughter cells were observed for the presence of newly synthesized green mkikGR fluorescence indicating the DTC fate. Subsequently, following the recovery and growth of the animals, the Dpd phenotype was assessed at the L3-L4 stages based on the positions of DTCs expressing green mkikGR fluorescence. (**C**) The Z1.p cells of *lin-17; egl-20 cwn-2; tkIs12* animals were laser-ablated at the late L1 stage. After the recovery and growth of the animals, either Z1.ap (left) or Z1.aa (right) cell was laser ablated. The migratory directions of DTCs derived from Z1.aa (left) or Z1.ap (right) were evaluated at the L4 stage by the positions of DTCs expressing *mig-24*::Venus. The animals were grown at 15°C (**A–B**) or 22.5°C (**C**). Scale bars indicate 2 µm. Source data is available (**Figure 4—source data 1**).

The online version of this article includes the following source data for figure 4:

**Source data 1.** Source data for abnormal migration of distal tip cells (DTCs) in *lin-17; cwn-2 egl-20* mutants in **Figure 4**.

stages, both left- and right-DTCs were present in the posterior region. These results strongly suggest that the Dpd phenotype is caused by the reversed migration of the DTC derived from Z1.

How are the migratory directions of DTCs controlled? Based on Z1 polarity reversal and the Dpd phenotype in *lin-17; egl-20 cwn-2* mutants, we initially examined the possibility that the direction of DTC migration correlates with the polarity of the DTC mother cell (DTC is a granddaughter of Z1/Z4) and that the Dpd phenotype is caused by the polarity reversal of the DTC mother. Since Z1 polarity is completely reversed in *lin-17; egl-20 cwn-2* at 15°C, the DTC is likely to be produced from Z1.p rather than Z1.a. To understand the relationship between Z1.p polarity and the directions of DTC migration derived from Z1.p, we first examined Z1.p polarity by GFP::POP-1 asymmetry between its daughter cells, and then observed DTC positions after the recovery and growth of the animals. In the observed triple mutants, Z1.p polarity was LH (*n*=5) and HL (*n*=5) polarity, indicating that Z1.pa and

Z1.pp, respectively, became DTCs. In both cases, the DTC migrated either anteriorly or posteriorly (*Figure 4A*), suggesting that the migratory direction is not correlated with Z1.p polarity.

The results suggest that DTCs produced at the position of Z1.pa or Z1.pp can migrate either anteriorly or posteriorly. To further confirm this, we identified DTCs after the Z1.p division using *lag-2*::GFP expression instead of GFP::POP-1. In the wild-type, the *lag-2* promoter::GFP is weakly expressed in the Z1/Z4 cell and its expression becomes stronger in Z1.aa/Z4.pp (DTC) soon after they are born (*Fujita et al., 2007*; *Henderson et al., 1994*). To unambiguously identify DTCs in *lin-17*; *egl-20 cwn-2* mutants, we expressed the photo-convertible fluorescent protein mKikGR under the *lag-2* promoter, converted it to red in Z1.p, and then, after the recovery and growth of the animals until the Z1.p division, we identified DTCs by newly synthesized green mKikGR. As shown in *Figure 4B*, when Z1.pp became a DTC, it migrated either anteriorly or posteriorly. Although we observed only one animal in which Z1.pa became a DTC, this animal showed the Dpd phenotype indicating posterior migration. Together with the experiments using GFP::POP-1 described above, the results demonstrated that when Z1.pa and Z1.pp become DTCs, they can migrate either anteriorly or posteriorly.

The results raised the possibility that ectopic positions of DTCs inside the gonad rather than at the distal ends in *lin-17*; *egl-20 cwn-2* mutants cause abnormal migration. If so, when Z1.aa (canonical DTC) becomes a DTC in the triple mutants, it should migrate anteriorly. To examine this, we isolated Z1.aa by sequential ablation of Z1.p and Z1.ap in *lin-17*; *egl-20 cwn-2* animals grown at 22.5°C. Since the reversal of Z1 polarity is partial at 22.5°C (*Figure 3B*), we expected that Z1.aa could become a DTC even in the triple mutants. However, all 10 animals we examined had one posterior gonadal arm, suggesting that Z1.aa rarely becomes DTC in the triple mutants (*Figure 4C*). In contrast, when Z1.ap was isolated by sequential ablation of Z1.p and Z1.aa, 3 out of 15 animals had a normal anterior arm, and 3 out of 15 showed the Dpd phenotype. Taken together, ectopic DTC produced at the positions of Z1.ap, Z1.pa, and Z1.pp migrate randomly, either anteriorly or posteriorly, at least in *lin-17*; *egl-20 cwn-2* animals.

## Ectopic positions of DTCs cause abnormal migration

These results suggest two possibilities explaining abnormal DTC migration in *lin-17*; *egl-20 cwn-2*. Ectopically positioned DTCs can migrate randomly irrespective of Wnt signaling mutations. Alternatively, migration of ectopic DTCs in addition to SGP polarity is regulated by Wnt signaling, and abnormal migration occurs only in the absence of some or all of *lin-17*, *egl-20*, or *cwn-2* functions. To distinguish these possibilities, we used a genetic background that produces ectopic DTCs without mutations in Wnt signaling genes. It was reported that RNAi of *cki-1* encoding a cyclin inhibitor causes ectopic DTCs (*Kostić et al., 2003*). However, the migratory direction of such ectopic DTCs was not examined. We confirmed that *cki-1(RNAi)* causes ectopic DTCs in addition to the loss of DTCs (*Figure 5A*). To determine the migratory direction of Z1-derived ectopic DTCs, we ablated Z4 in *cki-1(RNAi)* animals. In 7 out of 22 animals, we observed posteriorly migrated DTCs (*Figure 5B and C*). The results show that ectopic DTCs from Z1 can migrate either anteriorly or posteriorly irrespective of mutations in the Wnt signaling genes.

## DTCs can migrate independently of germ cells

It has been reported that DTCs are not self-propelled but rather pushed distally by proliferating germ cells (*Agarwal et al., 2022*). However, the posterior migration of DTCs produced from Z1 in *lin-17*; *egl-20 cwn*-2 mutants and Z4-ablated *cki-1(RNAi)* animals cannot be explained by the pushing mechanism of germ cells. Even in such animals, Z1-derived DTCs should be born at the anterior side of germ cells, and some of them migrate posteriorly. At the late L2 stage of *lin-17*; *egl-20 cwn-2* animals, differentiated DTCs expressing *mig-24*::Venus can be observed in the center of the gonad surrounded by germ cells expressing the germ cell marker *xnSi1* in addition to Z4-derived DTC at the posterior end of the gonad (*Figure 6A and B*). After the recovery and growth of such animals, we found animals with the Dpd phenotype (2/5 and 3/4 without and with a germ cell marker *xnSi1*, respectively), showing that Z1-derived DTCs migrate through but are not pushed by germ cells. In addition, among *mes-1* animals that lack germ cells, we found 84% of DTCs (*n*=90) were observed outside of the central region near the developing vulva at the L3-L4 stages, suggesting their distal migration to some extent (*Figure 6C–E*). We even found one animal (*n*=45) whose DTCs apparently underwent dorsal turn judged by the outline of the gonad marked by *lam-1*::mCherry (*Figure 6C*). The

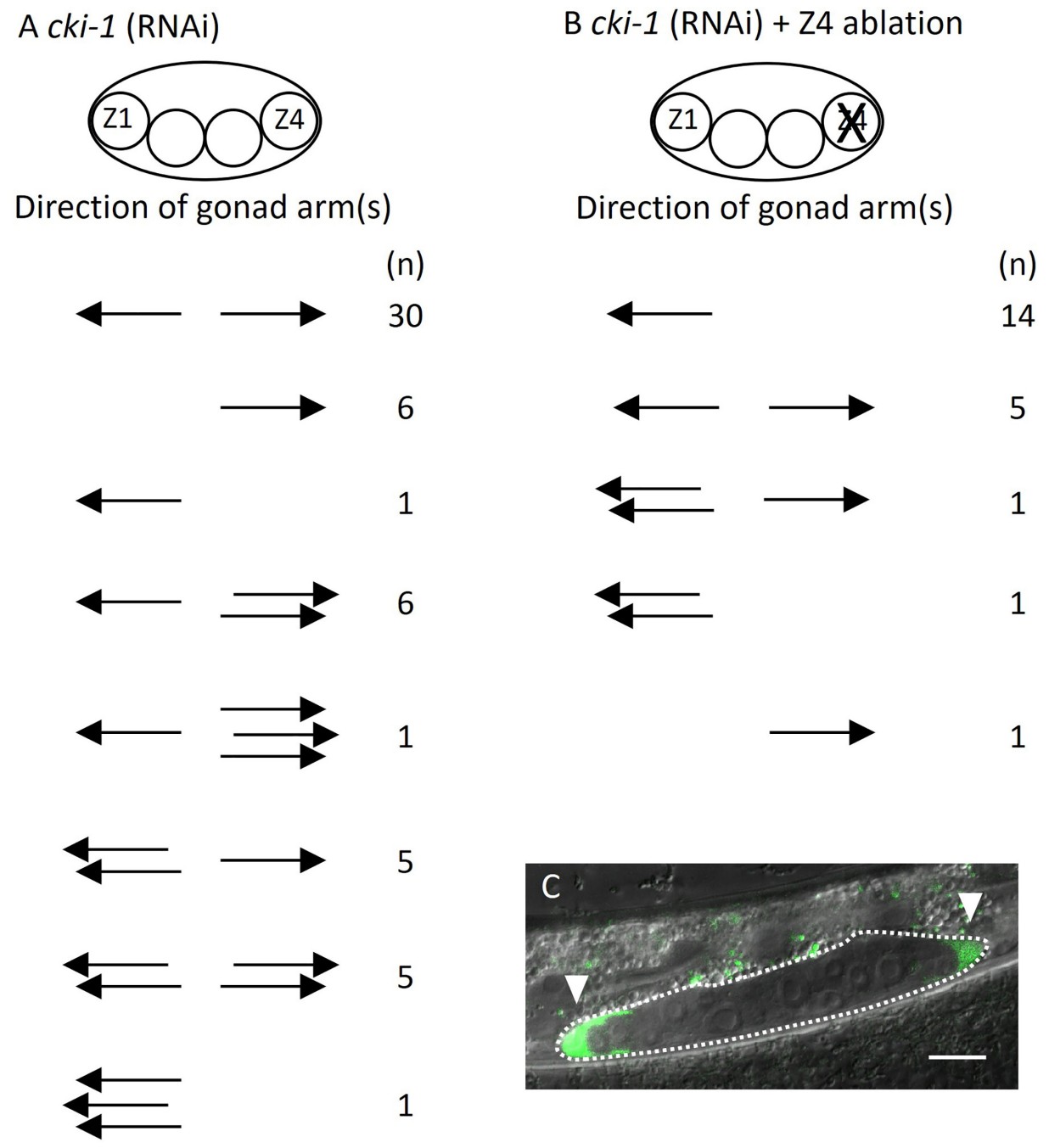

**Figure 5.** Ectopic positions of distal tip cells (DTCs) cause the Dpd phenotype. Migratory directions and numbers of DTCs in *cki-1(RNAi)* (**A**) and Z4 cell-ablated *cki-1(RNAi)* (**B**) animals grown at 15°C judged by *mig-24*::Venus at the L4 or young adult stages. Each arrow represents the migratory direction of an individual DTC. (**C**) An example of Z4 cell-ablated *cki-1(RNAi)* animal carrying anteriorly and posteriorly migrated DTCs (arrow heads) expressing *mig-24*::Venus. Scale bars indicate 10 μm.

results show that DTCs have the ability to migrate independently of germ cells. We noticed that some *mes-1* animals that lack germ cells have both of the two DTCs on the anterior side (2/45) or on the posterior side (5/45) (*Figure 6D*). This may be caused by polarity reversal of SGP (*Yamamoto et al., 2011*) or DTC mother cells placing DTCs at ectopic positions in *mes-1* mutants.

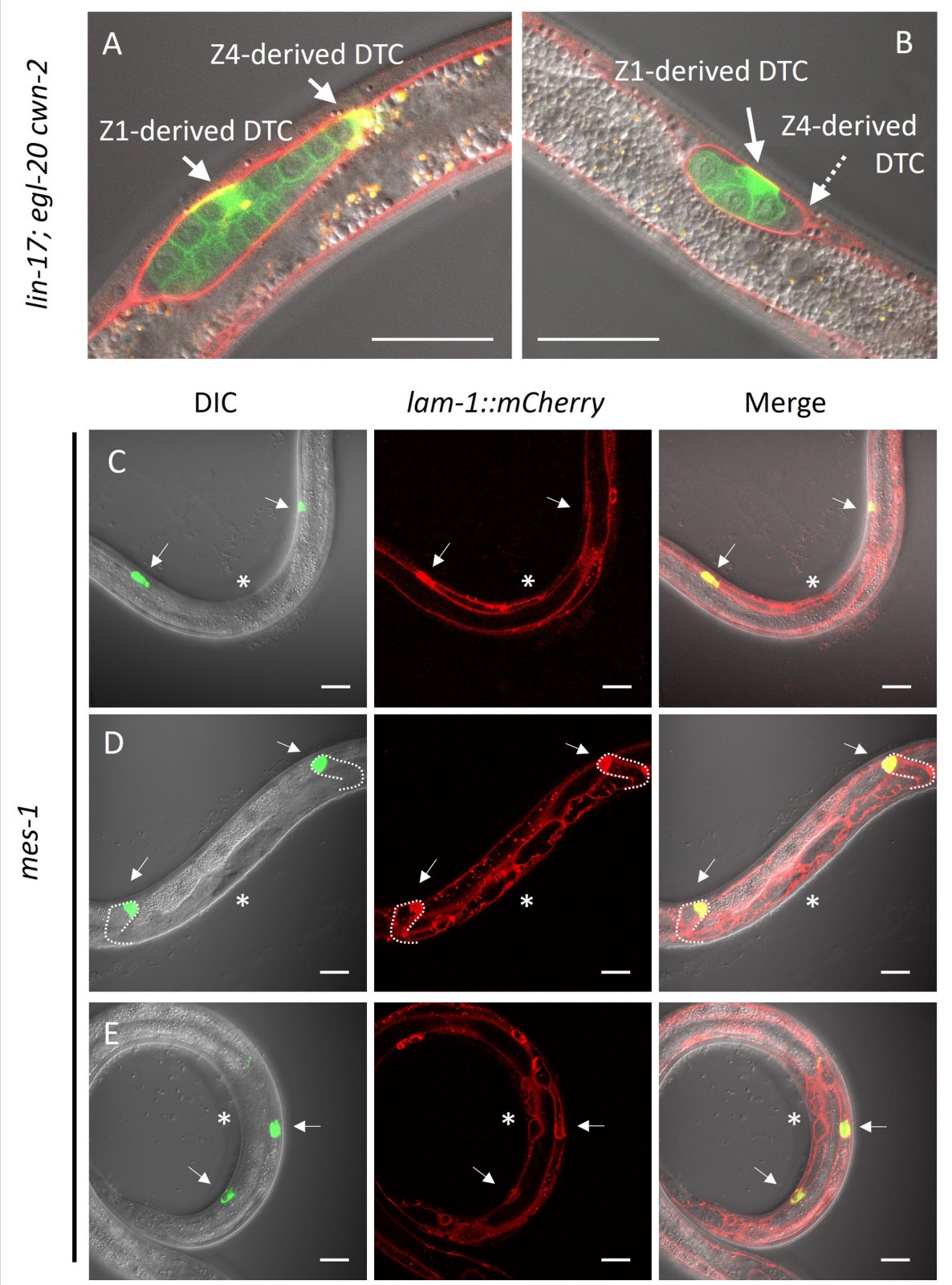

**Figure 6.** Germ cell-independent migration of distal tip cells (DTCs). (**A–B**) *lin-17; egl-20 cwn-2* and (**C–E**) germless *mes-1* animals carrying *lam-1*::mCherry (*qyIs127*) and *mex-5*::GFP::PH (*xnSi1*) and *mig-24*::Venus (*tkIs12* in (**A**) and *osEx283* in (**B–E**)) were grown at 15°C (**A–B**) or 22.5°C (**C–E**). The signals of *mex-5*::GFP::PH and *mig-24*::Venus can be distinguished by membrane and cytoplasmic fluorescence, respectively. Germless *mes-1* phenotype was confirmed by the absence of the *mex-5*::GFP::PH signal in the gonad. Merged images are shown in (**A–B**). In (**C–E**), merged images of

*Figure 6 continued on next page*

*Figure 6 continued*

DIC and green channels, red channel, and those of all channels are shown. In (**E**), merged images of two different focal planes are shown. Scale bars indicate 20 μm. Arrows indicate DTCs, and a dotted arrow in (**B**) indicates a DTC out of focus. Asterisks in (**C–E**) indicate positions of vulval investigation.

## Discussion

### Distinct functions of Wnts regulating SGP polarity

We have previously shown that the polarity of seam cells (V1-V5) is redundantly regulated by *cwn-1*, *cwn-2*, and *egl-20* (*Yamamoto et al., 2011*), indicating that all three of these Wnts induce the same polarity orientation (HL for high-low POP-1 concentration). In contrast, we show that these Wnts have distinct functions for SGP polarity at least in the *lin-17* mutant background (*Figure 7*). CWN-1 induces HL polarity for both Z1 and Z4. CWN-2 induces the opposite polarity; LH for Z1 and HL for Z4. EGL-20 inhibits the polarization of both Z1 and Z4. The distinct responses of Z1 and Z4 to CWN-2 appear to be a key for their opposite polarity orientation and mirror-symmetric gonadogenesis. Since these cells have asymmetry in terms of their contacts with germ cells on their proximal sides and those with the basement membrane on their distal sides, CWN-2 may permissively facilitate this asymmetry to control POP-1 localization. Alternatively, Z1 and Z4 may have intrinsic differences in gene expression, leading to distinct responses to CWN-2. The function of CWN-1 for SGP polarity appears to be the same as for seam cell polarity, inducing HL polarity in both cases. Since it was suggested that CWN-1 is a permissive signal for seam cell polarity (*Yamamoto et al., 2011*), it may also permissively regulate SGP polarity.

In contrast to the role of EGL-20 in seam cells, where it induces HL polarity, it inhibits cell polarization in SGPs. The loss of SGP polarity phenotype in the *lin-17; cwn-2* double mutant but not in the

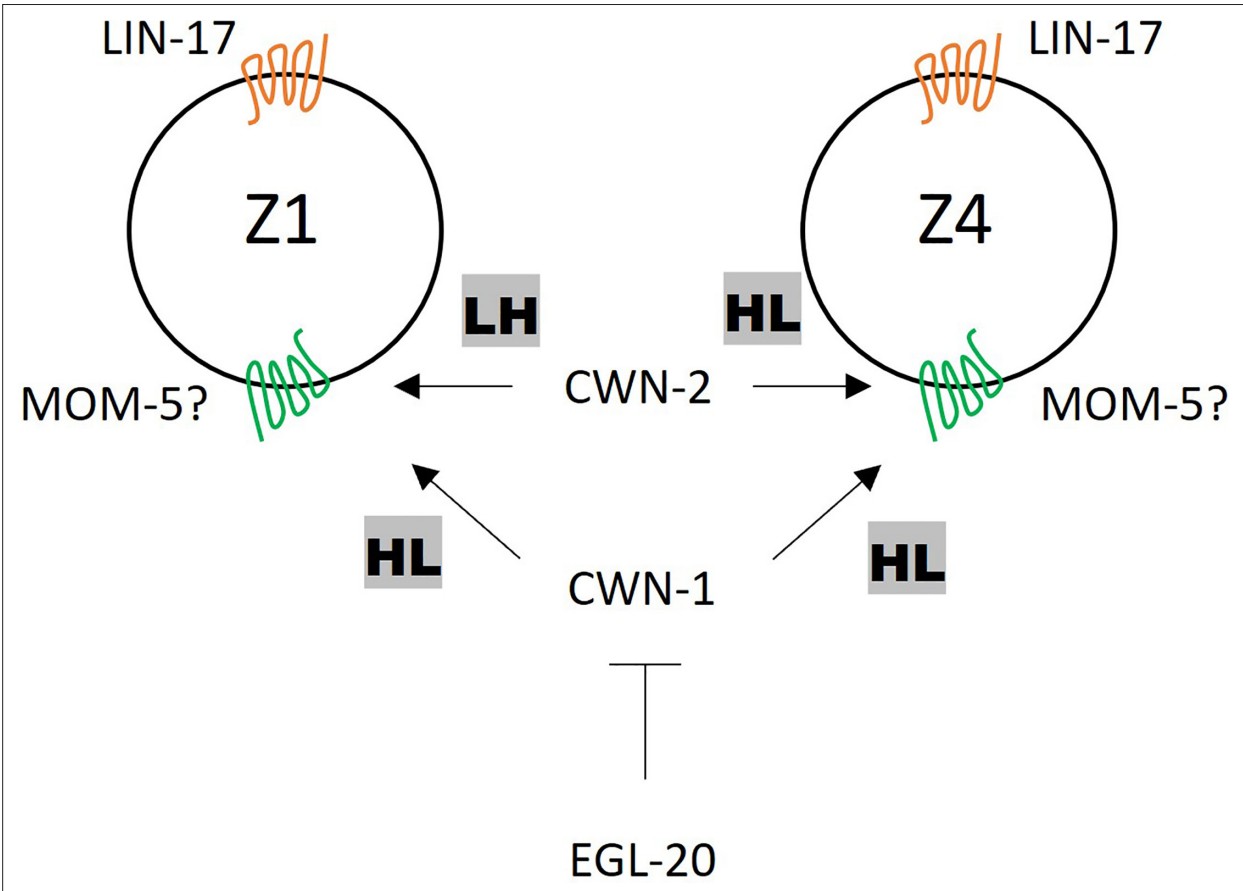

**Figure 7.** A model of somatic gonadal precursor (SGP) polarity regulation. LIN-17 regulates SGP polarity in a Wnt-independent manner. CWN-2 promotes LH and HL polarity in Z1 and Z4, respectively, while CWN-1 promotes HL polarity in both Z1 and Z4. EGL-20 inhibits the function of CWN-1. MOM-5 might serve as the receptor for both CWN-1 and CWN-2.

*lin-17; cwn-1 cwn-2* triple mutants was suppressed by further mutating *egl-20*. The recovered polarity in *lin-17; egl-20 cwn-2* depends on *cwn-1*, as *lin-17; cwn-1; egl-20 cwn-2* mutants showed a strong loss of polarity phenotype. However, *cwn-1* cannot efficiently polarize SGPs in *lin-17; cwn-2* mutants with normal *egl-20* function. Therefore, *egl-20* appears to inhibit the *cwn-1* function but not its expression (*Figure 3—figure supplement 2*). A similar antagonistic relationship between *cwn-1* and *egl-20* was reported for later DTC migration when DTCs turn to migrate centrally on the dorsal side (phase III) (*Levy-Strumpf and Culotti, 2014*). *egl-20* mutants show a reversal of phase III migration and this phenotype is suppressed by the *cwn-1* mutation. Further studies are necessary to elucidate the distinct functions of CWN-1, CWN-2, and EGL-20.

## Wnt-independent functions of Frizzled receptors

We have shown that *lin-17/Fzd* functions in a Wnt-independent manner to control SGP polarity, since the missing DTC phenotype of *lin-17; cwn-2* and *lin-17 mom-5* was completely rescued by ΔCRD-LIN-17. In addition, SGP polarity is normal in the quintuple Wnt mutant that has mutations in all the Wnt genes (*Yamamoto et al., 2011*). In seam cells, Wnt receptors including LIN-17/Fzd and MOM-5/Fzd appear to have Wnt-independent functions for cell polarization, as seam cells are still mostly polarized in the quintuple Wnt mutants, while they are strongly unpolarized in the triple receptor mutants (*lin-17 mom-5 cam-1/Ror*) (*Yamamoto et al., 2011*). In *Drosophila*, Fz/Fzd has been primarily considered to function Wnt-independently to coordinate PCP between neighboring cells (*Lawrence et al., 2007*), though Fz function can still be regulated by Wnt, as PCP orientation can be directed by ectopically expressed Wnt proteins (*Wu et al., 2013*).

In *Drosophila*, Fz regulates PCP by interacting with other PCP components including Van Gogh (Vang). In *C. elegans*, we found that *vang-1*/Vang does not appear to function with LIN-17/Fz, since most *vang-1* single mutants and *cwn-1 cwn-2 vang-1* triple mutants have two gonadal arms (215/216 and 58/58, respectively). As Fz interacts with Disheveled (DSH) in *Drosophila* PCP regulation, in *C. elegans*, the Disheveled homologs DSH-2 and MIG-5 regulate SGP polarity (*Phillips et al., 2007*). Therefore, LIN-17 might regulate the DSH homologs in a Wnt-independent manner.

## Control of DTC migration

We have shown that Z1-derived DTC can migrate posteriorly in *cki-1(RNAi)* as well as *lin-17; egl-20 cwn-2* animals, indicating that ectopically positioned DTCs can mis-migrate passing through germ cells irrespective of mutations in Wnt signaling genes. The results strongly suggest that DTCs at ectopic positions can self-migrate to the distal ends of the gonad, regardless of the pushing forces by germ cell proliferation. After reaching the distal ends, further distal migration is likely guided by germ cell proliferation and the degradation of the basement membrane at the distal sides (*Agarwal et al., 2022*). However, the self-migration of ectopic DTCs to the distal ends suggests that such migratory ability of DTCs may help their further distal migration.

While we were unable to observe migration of Z1.aa-derived DTC in the triple mutants, it is essential to note that Z1.aa-derived DTCs consistently migrate anteriorly in wild-type animals and likely in other genetic backgrounds, giving the absence of the Dpd phenotype in previous reports. Consequently, the regulation of SGP polarity by Wnt signaling plays a crucial role in ensuring normal DTC migration and mirror-symmetric organogenesis by positioning DTCs at the distal edge of the gonad.

## Materials and methods
### Strains

N2 Bristol was used as the wild-type strain (*Brenner, 1974*). The following mutations and transgenes were used: *cwn-1(ok546)* (deletion) (*Zinovyeva and Forrester, 2005*), *cwn-2(ok895)* (deletion) (*Zinovyeva and Forrester, 2005*), *egl-20(n585)* (missense behaving like null) (*Maloof et al., 1999*), *lin-17(n3091)* (nonsense) and *lin-17(n671)* (nonsense) (*Sawa et al., 1996*), *mig-1(e1787)* (nonsense) (*Pan et al., 2006*), *mom-5(ne12)* (nonsense) (*Rocheleau et al., 1997*), *cam-1(gm122)* (nonsense) (*Forrester et al., 1999*), *cfz-2(ok1201)* (deletion) (*Zinovyeva and Forrester, 2005*), *lin-18(e620)* (nonsense) (*Inoue et al., 2004*), *osIs113* (ΔCRD-LIN-17); *osEx395* (*ceh-22p*::CWN-1::VENUS) (*Yamamoto et al., 2011*), *osIs93* (*egl-20p*::CWN-2::VENUS), a spontaneous integration of *osEx402* (*Yamamoto et al., 2011*), *tkIs12* (*mig-24*::Venus) (*Tamai and Nishiwaki, 2007*), *qIs74* (*sys-1p*::GFP::POP-1)

(*Siegfried et al., 2004*), *qIs95* (VENUS::SYS-1) (*Phillips et al., 2007*), *qyIs127* (*lam-1*::mCherry) (*Ihara et al., 2011*), *xnSi1* (*mex-5*::GFP::PH) (*Chihara and Nance, 2012*). *cki-1(RNAi)* was performed as described previously (*Fujita et al., 2007*). *lin-17(os2)* was identified in a screen for the Psa (phasmid socket absent) phenotype (*Sawa et al., 2000*). *lin-17(os2)* and *lin-17(mn589)* (gifted by Mike Herman) carry mutations in the second and seventh cysteine residues of the CRD domain (C36Y and C104Y), respectively. *os2* and *mn589* exhibit 38% and 47% Psa phenotype (indicating T cell polarity defects), respectively, while *lin-17(n3091)* shows 95% Psa phenotype (*Goldstein et al., 2006*). *mes-1(bn7)* is a temperature-sensitive allele with higher penetrance of the germless phenotype at 25°C than at 15°C, and was grown at 22.5°C. The germless phenotype of *mes-1(bn7)* was observed by the absence of the *mex-5*::GFP::PH signal through direct observation of epifluorescence.

## Plasmid construction

ΔCRD-LIN-17 (pMM39) contains a 5 kb XhoI genomic fragment upstream of the *lin-17* start codon and a *lin-17* cDNA fused to Venus at its C-terminus in the pPD49.26 vector (gift of A Fire). The CRD domain was precisely deleted from the *lin-17* cDNA. *sys-1p*::GFP(NLS) (pSS20) contains the *sys-1* promoter from the GFP-POP-1 plasmid (pJK707) (*Siegfried et al., 2004*) inserted into pPD95.67 (gift of A Fire). *lag-2p*::NLS::mKikGR (pSN17.4) contains a 7.4 kb *lag-2* promoter fragment and NLS::m-KikGR coding sequence (codon-optimized). pMM39, pSS20, pSN17.4, and *mig-24*::Venus (*Tamai and Nishiwaki, 2007*) plasmids were injected into *unc-76(e911)* animals along with the Unc-76 rescuing plasmid (*Bloom and Horvitz, 1997*) to obtain *osEx576*, *osEx443*, *osEx509*, and *osEx283*, respectively. Subsequently, *osEx576* and *osEx509* were integrated by UV-irradiation to generate *osIs113* and *osIs168*, respectively.

## Quantification of POP-1 asymmetry in the Z1 and Z4 division

Localization of POP-1 and SYS-1 was observed using confocal microscopy (Zeiss LSM700) and signal intensities were quantified with ImageJ software. As SGP daughter cells are situated in distinct focal planes, the signal ratios of GFP::POP-1 (*sys-1p*::GFP::POP-1) [*qIs74*] between SGP daughter cells were normalized using the levels of *sys-1p*::GFP(NLS) [*osEx443*], which are considered to be the same between them. First, in wild-type animals with *osEx443* but not *qIs74*, signal intensities in the SGP daughter cells at focal planes showing maximum intensity in each cell were quantified. Distances (Z1.p-Z1.a or Z4.a-Z4.p) along the Z axis were also recorded. Ratios of signal intensities (Z1.p/Z1.a or Z4.a/Z4.p) on a logarithmic scale and distances were plotted on the Y axis and X axis, respectively.

Using these plots, a regression line was calculated:

$$y = -0.034x + 0.0148$$

To assess the variability of the predicted values, we calculated the predicted values of the regression line and their CI. First, the standard error (SE) of the residuals was determined, which was 0.0613. Next, given a sample size of 97, the degrees of freedom were 95. Based on these degrees of freedom, the *t*-value for a 95% CI was calculated. Finally, the upper ($e_{upper}$) and lower ($e_{lower}$) bounds of the the 95% CI for each predicted value $\hat{y}$ were calculated using the following formulas:

$$e_{upper} = \hat{y} + t \times SE$$
$$e_{lower} = \hat{y} - t \times SE$$

The deviations of the upper and lower bounds of the 95% confidence interval were defined as follows:

$$v_{upper} = e_{upper} - \hat{y}$$
$$v_{lower} = e_{lower} - \hat{y}$$

Subsequently, distances (Z1.p-Z1.a or Z4.a-Z4.p) along the Z axis as well as fluorescent intensities in both wild-type and mutant animals with *sys-1p*::GFP::POP-1 (*qIs74*) were recorded, and the distances were used as the x values in the regression equation to calculate theoretical ratios of fluorescence expressed equally between SGP daughter cells. Then, the observed fluorescence ratios of *sys-1p*::GFP::POP-1 on a logarithmic scale were adjusted by subtracting these theoretical values calculated from the distances (*ri*).

$$ri = yi - \hat{y}i$$

Finally, the absolute and signed values of the results (absolute and signed differences) were used to evaluate extent and orientation of SGP polarity, respectively. To evaluate the absolute differences, cases where the residuals were greater than $v_{upper}$ were assessed as exhibiting polarity formation ($ri > v_{upper}$). Conversely, for the signed differences, cases where the residuals were greater than $v_{upper}$ were defined as having normal polarity($ri > v_{upper}$), and cases where the residuals were less than $v_{lower}$ were defined as having reversed polarity ($ri < v_{lower}$).

The violin plot was created using the Seaborn library in Python. For absolute differences ($|ri|$ values were used), the parameter cut=0 was utilized in violin plots to truncate the kernel density estimation at the minimum and maximum data points, ensuring that the density tails did not extend beyond the actual data range. In contrast, for signed differences ($ri$ values were used), the parameter cut=2 was used, allowing the kernel density estimation to extend beyond the observed data range, providing a smoother and more comprehensive visualization of the data distribution. Additionally, to maintain a consistent bandwidth for kernel density estimation in the violin plots, the parameter $bw$=0.2 was set for all plots. $v_{upper}$ and $v_{lower}$ were used as the 95% CI on the violin plot. We used the Student's t-test to compare the differences in fluorescence intensities between the two groups of interest. The statistical analysis was performed using JMP9 software (SAS, USA).

## Scoring the absence and migration of DTCs

The presence, absence, and positions of DTCs were determined by direct observation of the fluorescence of *mig-24*::Venus using epifluorescent microscopy. For animals lacking *mig-24*::Venus, the assessment was made based on gonad arms under Nomarski microscopy for animals that do not have *mig-24*::Venus. These observation were performed at the L4 or young adult stages. Initial DTC migratory direction was judged by considering the position and cup shape of the DTCs. Anterior and posterior positions of DTCs indicated initial anterior and posterior migration, respectively. When DTCs were observed in the center, their migratory direction (which is opposite to the initial migratory direction) was judged by assessing their cup shape.

## Cell ablation and photoconversion

The animals were immobilized on slide glasses using 10 mM sodium azide. Laser ablation was performed by a MicroPoint system (Photonic Instruments) equipped with a 2 mW pulsed nitrogen laser (model VL-337; Laser Science Inc) exciting Coumarin 440 dye. Photoconversion was performed through irradiation with 405 nm laser on Zeiss LSM510 confocal microscope.

## Acknowledgements

We thank Judith Kimble for pJK707 and communicating unpublished results, Mike Herman for *lin-17(mn589)*, Katsuyuki Tamai and Kiyoji Nishiwaki for *tkIs12*, Sohei Nakayama and Misato Matsuo for technical helps, Takefumi Negishi for comments on the manuscript. Some of the strains were provided by the CGC, which is funded by NIH Office of Research Infrastructure Programs (P40 OD010440). This work was supported by a JSPS KAKENHI grant (JP16H04797 to HS), a grant from Takeda Science Foundation (to HS) and NIG-JOINT (88A2024 to SS).

## Additional information

### Funding

| Funder | Grant reference number | Author |
|---|---|---|
| Japan Society for the Promotion of Science | JP16H04797 | Hitoshi Sawa |
| Takeda Science Foundation | | Hitoshi Sawa |
| NIG-JOINT | 88A2024 | Shuhei So |

| Funder | Grant reference number | Author |
|--------|------------------------|--------|

The funders had no role in study design, data collection and interpretation, or the decision to submit the work for publication.

## Author contributions
Shuhei So, Conceptualization, Resources, Funding acquisition, Investigation, Writing – review and editing; Masayo Asakawa, Resources; Hitoshi Sawa, Conceptualization, Supervision, Funding acquisition, Writing – original draft, Writing – review and editing

## Author ORCIDs
Shuhei So (ID) http://orcid.org/0000-0002-1730-6693
Hitoshi Sawa (ID) https://orcid.org/0000-0001-9774-7976

## Decision letter and Author response
Decision letter https://doi.org/10.7554/eLife.103035.sa1
Author response https://doi.org/10.7554/eLife.103035.sa2

## Additional files

### Supplementary files
• Supplementary file 1. List of strains used for experiments.
• MDAR checklist

### Data availability
All data generated or analysed during this study are included in the manuscript and supporting files; source data files have been provided for Figures 3 and 4.

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
