## [Editor Report]

The Wnt signaling pathway controls cell polarity, although the role of Wnts themselves remains controversial. This important study addresses the involvement of the Wnt pathway in the specification and migration of *C. elegans* DTCs and shows how they are symmetrically polarized. The paper presents compelling evidence that will be of interest for understanding a striking example of cell polarity and, for the *C. elegans* community, to show that DTCs can migrate independently of the germline.

---

## [Decision Letter]

[Editors' note: this paper was reviewed by Review Commons.]

---

## [Author Response]

1. Point-by-point description of the revisionsReviewer 1:I recommend combining Figures 1 and 3 either before or after the results shown in Figure 2, so the reader's expectation for quantification is immediately satisfied.

Thank you for your suggestion. In the revised manuscript, images of GFP::POP-1 in compound mutants are moved to Figure 3. The schematic diagram of the gonad (previously Figure 1A) and GFP::POP-1 images in wild type are kept in Figure 1, as they are described in Introduction.

Major comments:Delving into the figure legend of Figure 3 and the normalization procedure described in the Methods "Quantification of POP-1 asymmetry in the Z1 and Z4 division" raised concerns. The method therein described is overly complicated but also neglects background subtraction. My first question about this method: what range of distances between daughters is measured in Z? These distances are not discussed in absolute terms, and this is important for our understanding of how much correction for tissue depth might be necessary, as L1s are very thin.To check my understanding, the authors use as a control a nuclear-localized GFP driven in the somatic gonad precursors in otherwise wild-type worms by the sys^-1^ promoter. They observe that the regression on a log scale of anterior:posterior (and vice versa) Z1 and Z4 daughter fluorescence over the distance between the daughters in the Z plane is fit by y = −0.034x + 0.0148, which is practically a slope of 0 and an intercept of 0. This means that they observed an ~1:1 ratio (as log(1)=0) of fluorescence in the anterior and posterior daughters of otherwise wild-type worms, at least across the range of very small X values of relevant distances between daughters (again, the relevant range of distances really matters and should be presented), making the normalization seem unnecessary.

Normalization is essential to compare POP-1 signals between daughter cells since the signal intensities depend on the depth of cells. Depth differences between SGP daughter cells range from 0 to 7.5 micrometers. For example, when we input the maximum difference (7.5) into our correction equation y = −0.034x + 0.0148 (the logarithmically transformed linear regression equation), we get:

y = −0.034 * 7.5 + 0.0148 = -0.2402

To interpret this on the original scale, we apply the inverse logarithmic transformation:

10^(-0.2402) ≈0.575

This result indicates that even if GFP::POP-1 expression is the same in both cells, the depth difference alone can cause approximately a 1.74-fold (1/0. 575) difference in fluorescence intensity.

Similarly, if we use a median value of 3.5 micrometers as the depth difference, we get: y = -0.1042. After the inverse logarithmic transformation, this corresponds to a 0.787 or 1.27 (1/0.787) fold difference in fluorescence intensity.

Without normalization, we risk misinterpreting such differences in expression levels when in reality the expression is the same. Conversely, actual differences in GFP::POP-1 signal could be masked or overestimated due to the depth effect.

In the revised manuscript, examples of depth differences between SGP daughters are shown in Figure 2S which is added in response to the comment of reviewer 2, asking images of lin-17 mom-5 animals.

In the revised manuscript, we explained the depth effects in the legend of Figure 3 as follows.

“Since SGP daughter cells are often present at distinct focal planes, we normalized the depth effects on fluorescence intensities (see Materials and methods for details) for the quantification shown in (B). The images in (A) and (C) are from animals with SGP daughters at similar depths.”

Then based on this regression and 95% CI, the authors predict values that reflect true equivalence of fluorescence of POP-1::GFP in the two SGP daughters, compare the observed values to these predictions, and ultimately display in violin plots these differences of observed and expected. Correct?

Yes, your understanding is correct.

Is this complicated treatment the only way to detect differences in polarity of anterior and posterior daughters of Z1 and Z4? What happens if the authors measure GFP::POP-1 and calculate the following?Z1p(MGV – background control from same focal plane)Z1a(MGV – background control from same focal plane)If this straightforward analysis shows asymmetric signal in the control that is made symmetrical or reversed in the mutants, the hypothesis would seem to be supported with a much more straightforward method. Samples could be analyzed separately in two bins by worm body position, which affects which cell is superficial in the sample. As it is, the Figure 3 Y axis label is hard to interpret without reading the methods at length, diminishing its impact.

Thank you for the suggestion. Your suggested calculation would be simple if we could assume that control signals (sys^-1^p::GFP::NLS or sys^-1^p::GFP::POP-1 in the same wild-type cell) on the same focal plane are the same among animals. However, since there are apparent variations in expression levels among individuals, your suggested method is not appropriate for evaluating differences in sys^-1^p::GFP::POP-1 intensities between the SGP daughter cells of the same animal.

Missing control: The sys^-1^ promoter-driven NLS-tagged fluorescent protein as a control to compare to the GFP::POP-1 is analyzed only in the wild-type, and apparently not in the mutants under consideration. Phillips et al. (2007) show that sys^-1^p transcriptional activity is equivalent between the SGP daughters in wild-type worms, but neither those results nor the method of normalizing to a sys^-1^p::GFP::NLS signal in this paper address the question of whether sys^-1^ promoter activity is equivalent in these cells in mutants upstream in the Wnt pathway. If the current method of normalization is to be used, it seems important to normalize to the sys^-1^p::GFP::NLS regression in each mutant background.

Thank you for your suggestion. We used sys^-1^p::GFP::NLS as a control to normalize depth effects, which should be the same across all genotypes because the GFP molecules in SGPs should be equally distributed between SGP daughter cells, not because sys^-1^ promoter activities are similar among them. Since SGP daughters divide within a short time (about 2 hours), it is likely that the fluorescence of newly synthesized GFP (maturation time of about 1 hour) in SGP daughters is neglectable compared to GFP inherited from the SGP cells. Similarly, sys^-1^p::GFP::POP-1 signals in SGP daughters reflect the distribution of GFP::POP-1 from SGPs rather than the transcriptional activities of the sys^-1^ promoter in the daughter cells. sys^-1^p::GFP::POP-1 or sys^-1^p::GFP::SYS^-1^ has been widely used to evaluate polarity of asymmetric divisions in a number of studies, none of which consider transcriptional differences of the sys^-1^ promoter in the daughter cells.

1. How was lin-17(mn589) generated? if this is the first report of this allele, full information on what the lesion is and how it was derived should to be reported.

Thank you for your question regarding the lin-17(mn589) allele. We would like to point out that the information about this allele is provided in the Methods section of the original manuscript as follows.

“lin-17(mn589) (gifted by Mike Herman) carries a mutation in the seventh cysteine residue of the CRD domain (C104Y). mn589 exhibits 47% Psa phenotype (indicating T cell polarity defects).”

2. The methods section lacks a description of how the mes^-1^ experiments were done, in terms of timing, duration, and temperature; mes^-1^(bn7) is a temperature sensitive allele.

Thank you for pointing out the lack of detailed methodology for the mes^-1^ experiments. The germless phenotype of mes^-1^ mutants is partial even at high temperatures. We have not performed temperature shifts to observe the phenotype. As per your suggestion, we added the following text to the Strains section:

"mes^-1^(bn7) is a temperature-sensitive allele with higher penetrance of the germless phenotype at 25°C than at 15°C, and was grown at 22.5°C. The germless phenotype of mes^-1^(bn7) was observed by the absence of the mex-5::GFP::PH signal through direct observation of epifluorescence."

Minor comments1. The paper lacks a discussion of precedent in the literature for Wnt-independent Frizzled activity; this is a major finding that is being undersold in the current version of the manuscript.

Thank you very much for appreciating out finding. We have added the following paragraph to the Discussion section:

“Wnt-independent functions of Frizzled receptors

We have shown that lin-17/Fzd functions in a Wnt-independent manner to control SGP polarity, since the missing DTC phenotype of lin-17; cwn-2 and lin-17 mom-5 was completely rescued by ΔCRD-LIN-17. In addition, SGP polarity is normal in the quintuple Wnt mutant that has mutations in all the Wnt genes (Yamamoto et al., 2011). In seam cells, Wnt receptors including LIN-17/Fzd and MOM-5/Fzd appear to have Wnt-independent functions for cell polarization, as seam cells are still mostly polarized in the quintuple Wnt mutants, while they are strongly unpolarized in the triple receptor mutants (lin-17 mom-5 cam-1/Ror) (Yamamoto et al., 2011). In *Drosophila*, Fz/Fzd has been primarily considered to function Wnt-independently to coordinate planar cell polarity (PCP) between neighboring cells (Lawrence et al., 2007), though Fz function can still be regulated by Wnt, as PCP orientation can be directed by ectopically expressed Wnt proteins (Wu et al., 2013).

In *Drosophila*, Fz regulates PCP by interacting with other PCP components including Van Gogh (Vang). In *C. elegans*, we found that vang-1/Vang does not appear to function with LIN-17/Fz, since most vang-1 single mutants and cwn-1 cwn-2 vang-1 triple mutants have two gonadal arms (215/216 and 58/58, respectively). As Fz interacts with Disheveled (DSH) in *Drosophila* PCP regulation, in *C. elegans*, the Disheveled homologs DSH-2 and MIG-5 regulate SGP polarity (Phillips et al., 2007). Therefore, LIN-17 might regulate the DSH homologs in a Wnt-independent manner. “

Added Reference:

1. Lawrence PA, Struhl G, Casal J. (2007). Planar cell polarity: one or two pathways? Nat Rev Genet. 8, 555-563.

2. Wu, J., Roman, A.C., Carvajal-Gonzalez, J.M., & Mlodzik, M. (2013). Wg and Wnt4 provide long-range directional input to planar cell polarity orientation in *Drosophila*. Nature Cell Biology, 15(9), 1045-1055.

2. Important: I think "Figure 6 Germ cell independent migration of germ cells" title is a typo; should be "Germ cell independent migration of DTCs"

Thank you for pointing out the typo. We corrected it in the revised manuscript.

3. This is a very important experiment! I think a greater description of the mes^-1^ phenotype would be helpful, since loss of germline was not 100% penetrant in mes^-1^(bn7) hermaphrodites in Strome et al., 1995. The legend says "Germless mes^-1^ phenotype was confirmed by the absence of the mex-5::GFP::PH signal in the gonad." Consider adding a few sentences to the results (or methods, from which the mes^-1^ experiments are currently missing) describing that only mes^-1^ animals that lacked germline fluorescence were analyzed for DTC migration.

Thank you for providing the context. To address the concerns, we made the following changes to our manuscript:

1. In the Results section, we revised the sentence "We found that 84% of DTCs (n = 90) in germless mes^-1^ animals…" to "Among mes^-1^ animals that lack germ cells, we found 84% of DTCs (n = 90)…".

2. We also modified the sentence "We noticed that some germless mes^-1^ animals…" to "We noticed that some mes^-1^ animals that lack germ cells…".

4. Please correct "secreting the Notch ligand LAG-2" this is a membrane-bound, not secreted ligand

Thank you for your comment. In the revised manuscript, we modified the relevant sentence in the Introduction section as follows:

“Firstly, DTCs function as niche cells for germline stem cells, inhibiting their entry into meiosis by expressing the Notch ligand LAG-2 (Henderson et al., 1994).”

5. Figure 1. The qualitative loss of polarity would be better depicted with a grayscale image instead of green-on-black.

Thank you for your suggestion. The GFP::POP-1 images are raw images of the green channel of the confocal microscopy. We believe that SGP polarity is clearly depicted by them.

6. Figure 3 the presentation of these violin plots is confusing. The central text that reads "normal polarity, loss of polarity, reversed polarity" with arrows looks like a second Y axis label attached to the Z4 plot. I recommend rearranging. Consider shading the top, bottom, and central regions and explaining the meaning of the shading in the legend.

Thank you for your suggestions regarding the presentation of Figure 3. In response to your feedback, we have made the following modifications:

First, we moved the text and arrows from the center to the right side of the figure, creating a clearer layout. As you recommended, we applied shading to the top, bottom, and central regions of the violin plots. Additionally, to explain the meaning of the shading, we added a new explanation to the figure legend. Specifically, we included the following text:

"Values within the 95% CI (between the red lines; light green regions) indicate symmetric localization. Values below the lower red line (light blue regions) indicate reversed localization, while values above the upper red line (light red regions) indicate normal localization.”

We applied the same modification to Supplemental Figure 1.

Reviewer 2:Major comments1. Are the effects of combining the different Wnts with the lin-17 allele specific to the n3091 allele? It would be important to test another allele, for example the sy277 allele has a similar phenotype and is available at CGC. A null would be even better if it is viable. Alternatively, lin-17(RNAi) could instead be used if efficient enough. This is important since the n3091 allele could differentially alter the binding to the various Wnts, resulting in their distinct phenotypes in that background. However, these distinct phenotypes may not be relevant in a wild-type context.

Thank you for your insightful comment. The lin-17(n3091) allele contains a nonsense mutation at the 35th codon, located between the second and third cysteine residues in the CRD domain (Wnt binding domain) (Sawa et al. 1996). Therefore, it is highly unlikely that the N-terminal protein of 34 amino acids produced in lin-17(n3091) can bind to Wnts. In the revised manuscript, we added the missing-DTC phenotype of lin-17(n671) cwn-2 animals, which show a similar phenotype to lin-17(n3091) cwn-2. n671 is a reference allele in WormBase and has a nonsense mutation. Although sy277 has a deletion in the N-terminal region, its phenotype is weaker than that of n3091 and n671 (Sawa et al. 1996).

In the revised manuscript, we described lin-17(n671) cwn-2, in the Table 1, Table S1 and added the following sentence.

“We observed a similar phenotype in lin-17(n671); cwn-2 double mutants, confirming that this genetic interaction is not allele-specific.”

2. In the lin-17; mom-5 double mutant which lacks DTCs, are Z1 and Z4 there but they do not express DTC markers, or are they never born? A lineage analysis should be presented. Also, are Z2 and Z3 still there on their own? Please show images.

Thank you for your comments. We quantified sys^-1^p::GFP::POP-1 signals in Z1 and Z4 daughter cells of lin-17 mom-5 and have not observed any animals lacking Z1, Z4 or germ cells. In the revised manuscript, as Figure S2, we added images of sys^-1^p::GFP::POP-1 localizations in SGP daughters, along with germ cells in lin-17 mom-5 as well as in lin-17 cwn-1 egl-20 cwn-2, both of which were not shown in the original manuscript. In response to Reviewer 1’s comment, we also included examples of depth effects on fluorescence intensities in Figure S2 with images of different focal planes.

Figure S2 is quoted it at the end of the following sentence.

“Then, we quantified the ratios (on a logarithmic scale) of sys^-1^p::GFP::POP-1 signal intensities proximal to distal daughter cells in various genotypes (Figure 3A and Figure S2).”

The loss of polarity phenotype of lin-17 mom-5 has been described in Phillips et al. We missed to cite this in the original manuscript. We added the citation in the revised manuscript.

“These asymmetries were strongly disrupted and weakly affected in lin-17 mom-5 double and lin-17 single mutants, respectively, as described previously (Phillips et al., 2007; Siegfried et al., 2004).”

Minor comments1. The term "mirror-symmetry" is redundant. Consider using "symmetry" or "symmetrical polarity".

As noted in the cross-comment by Reviewer 1, we believe that "mirror-symmetry" is the appropriate term.

We think that “symmetry” implies the same lineage, whereas the relationship between the Z1 and Z4 lineages is not. “Mirror symmetry” was also used in Herman & Horvitz (1994) to describe the defect in the F lineage in lin-44/Wnt mutants as follows.

“we observed division patterns that were mirror symmetric to those of the wild type (Figure 2). One plausible explanation is that the polarity of the first asymmetric cell division was reversed, causing the polarities of all subsequent asymmetric cell divisions also to be reversed.”

2. "… they are permissively pushed distally by germ cells while proliferating" is confusing as it is unclear what proliferating cell you are referring to – germ cells or the DTC? proliferating? sense. Replace by: "they are pushed distally by proliferating germ cells"

Thank you for your helpful comment. We agree with your suggestion and modify the sentence as follows:

Original: "… they are permissively pushed distally by germ cells while proliferating" Revised: "… they are pushed distally by proliferating germ cells"

3. Figure 2 is cited in the text before Figure 1.

Thank you for pointing this out. Figure1 is mentioned in the Introduction before Figure 2 is referenced in the Result section in the original manuscript. We think the reviewer might be confused, as the POP-1 localization defect was shown in Figure 1. In response to the reviewer 1’s comment, we moved the POP-1 localization images of the compound mutants to Figure 3. In addition, we noticed that in the original manuscript, Figure 1B was mentioned before Figure 1A in the Introduction. Therefore, we have modified the sentences in the Introduction.

The original sentence was:

"In the gonad, at the L1 stage, somatic gonadal precursor cells (SGPs), Z1 and Z4 have LH and HL polarity, respectively (Siegfried et al., 2004) (Figure 1B). This mirror-symmetric polarity creates their mirror-symmetric lineages producing distal tip cells (DTCs) from the distal daughters (Z1.a and Z4.p) (Figure 1B)."

The revised sentence now reads:

"In the gonad, at the L1 stage, somatic gonadal precursor cells (SGPs), Z1 and Z4 have LH and HL polarity, respectively, creating their mirror-symmetric lineages producing distal tip cells (DTCs) from the distal daughters (Z1.a and Z4.p) (Siegfried et al., 2004) (Figure 1A and 1B)."

4. The results also suggest that MOM-5/Frizzled might be the receptor for Wnts regulating DTC production, as lin-17 mom-5 double mutants completely lack DTCs." Table 1 results rather suggest that lin-17 and mom-5 are the two frizzled receptor involved in DTC specification and that they are largely redundant.

As the reviewer noted, lin-17 and mom-5 function redundantly in DTC specification (SGP polarization). However, their functions are clearly different in terms of genetic interactions with Wnt genes (e.g. lin-17 cwn-2 but not mom-5 cwn-2 show the DTC-missing phenotype). We propose that MOM-5 but not LIN-17 functions as a receptor for Wnts.